# Intramolecular charge-transfer enhances energy transfer efficiency in carotenoid-reconstituted light-harvesting 1 complex of purple photosynthetic bacteria

Nao Yukihira[1,5], Chiasa Uragami[1,5], Kota Horiuchi[1], Daisuke Kosumi [2], Alastair T. Gardiner [3], Richard J. Cogdell [4] & Hideki Hashimoto [1✉]

In bacterial photosynthesis, the excitation energy transfer (EET) from carotenoids to bacterio-chlorophyll *a* has a significant impact on the overall efficiency of the primary photosynthetic process. This efficiency can be enhanced when the involved carotenoid has intramolecular charge-transfer (ICT) character, as found in light-harvesting systems of marine alga and diatoms. Here, we provide insights into the significance of ICT excited states following the incorporation of a higher plant carotenoid, β-apo-8'-carotenal, into the carotenoidless light-harvesting 1 (LH1) complex of the purple photosynthetic bacterium *Rhodospirillum rubrum* strain G9+. β-apo-8'-carotenal generates the ICT excited state in the reconstituted LH1 complex, achieving an efficiency of EET of up to 79%, which exceeds that found in the wild-type LH1 complex.

[1] Department of Applied Chemistry for Environment, Graduate School of Science and Technology, Kwansei Gakuin University, 1 Gakuen-Uegahara Sanda Hyogo 669-1330, Japan. [2] Institute of Industrial Nanomaterials, Kumamoto University, 2-39-1 Kurokami Chuou-ku Kumamoto 860-8555, Japan. [3] Laboratory of Anoxygenic Phototrophs, Institute of Microbiology, Czech Academy of Sciences, 379 81 Třeboň, Czech Republic. [4] Institute of Molecular, Cell and Systems Biology, College of Medical, Veterinary and Life Sciences, University of Glasgow, Glasgow G12 8QQ, Scotland, UK. [5]These authors contributed equally: Nao Yukihira, Chiasa Uragami. ✉email: hideki-hassy@kwansei.ac.jp

The light-harvesting system found in purple photosynthetic bacteria[1] provides one of the best understood examples of photosynthetic light-harvesting. A great number of studies have been carried out on it to investigate the mechanisms of the energy transfer during light harvesting[2,3]. Learning lessons from photosynthetic light-harvesting can provide important information about how to construct efficient and robust light-harvesting systems that can be utilized in 'artificial photosynthesis'[4].

There are two types of light-harvesting (LH) pigment-protein complexes in purple photosynthetic bacteria. The core light-harvesting 1 (LH1) complex surrounds the reaction center (RC), and the peripheral light-harvesting 2 (LH2) complexes surround the LH1-RC core complex[5]. Solar energy that is captured by the LH2 complex is transferred to LH1 and then to the RC with a quantum efficiency of almost unity. A recent Cryo-EM study has described the detailed molecular architecture of the LH1-RC complex from the purple photosynthetic bacterium *Rhodospirillum* (*Rsp.*) *rubrum* strain S1[6]. This LH1 complex consists of 16 equally spaced α-β subunits forming a ring structure in which the light-absorbing pigments, carotenoid (spirilloxanthin), and bacteriochlorophyll *a* (Bchl *a*) molecules are arranged. Bchl *a* has a Soret absorption band in the ultraviolet spectral region, a $Q_x$ absorption band in the red spectral region, and a $Q_y$ absorption band in the near-infrared spectral region. BChl *a* does not absorb strongly in the blue-green spectral region of light, where the intensity of solar radiation distribution is the strongest. Carotenoids have their main absorption bands in this blue-green spectral region, so they can provide an important boost to light harvesting[7].

Carotenoids found in light-harvesting complexes typically consist of a long polyene backbone that has $N$ conjugated C=C bonds, where $N$ varies between 9 and 13[8]. There is a close relationship between the number of conjugated double bonds $N$ and the efficiency of excitation energy transfer (EET). The wavelength of light absorbed by carotenoids depends on $N$[9]. As $N$ increases, the EET from carotenoid to Bchl *a* tends to become less efficient[10,11] (see Fig. 1). About 90% of the carotenoids that are bound to the native LH1 complex from *Rsp. rubrum* strain S1 are spirilloxanthin ($N = 13$), while other minor carotenoids such as rhodovibrin ($N = 12$), anhydrorhodovibrin ($N = 12$), rhodopin ($N = 11$), and lycopene ($N = 11$) are also found, depending on the growth conditions[8,12]. In the LH1 complex of *Rsp. rubrum* strain S1, the efficiency of EET from carotenoids to Bchl *a* is 27%. Previous

studies have shown that the EET process from carotenoids to Bchl *a* is dominated by energy transfer from the carotenoid $S_2$ state to the $Q_x$ state of B880 Bchl *a* in the native LH1 complex from *Rsp. rubrum* strain S1[13–17]. On the other hand, Nakagawa et al. reported that the EET efficiency from carotenoid to Bchl *a* can vary from 28 to 65% in the carotenoid-reconstituted (for $N = 13–10$) LH1 complex from *Rsp. rubrum* strain S1[18], while Akahane et al. reported that it varies from 36 to 78% in the carotenoid-reconstituted (for $N = 13–9$) LH1 complex from *Rsp. rubrum* strain G9+, a carotenoidless strain[19]. For the carotenoids with shorter conjugation lengths ($N = 10$ and 9), the EET from the $S_1$ state of carotenoids becomes also active[16], which contributes to an increase in the efficiency of carotenoid to Bchl *a* EET. Furthermore, it can be seen that the efficiency of EET from carotenoid to Bchl *a* is proportional to the inverse of the number of conjugated double bonds ($1/N$) (see Fig. 1)[20].

Even though the longer conjugated carotenoids have a lower efficiency of EET they have a superior function in photoprotection[21]. The photoprotective function of carotenoids is mediated by the transfer of triplet-triplet excitation energy from the $T_1$ state of Bchl *a* to the carotenoid triplet state. Excess light exposure to the LH complexes enhances the production of the $T_1$ Bchl *a*, which reacts with oxygen to produce singlet oxygen. By scavenging the $T_1$ Bchl *a* with carotenoid, the generation of singlet oxygen is suppressed, which protects the LH system.

Carotenoids that contain a carbonyl group coupled to their polyene backbone are known to generate an intramolecular charge-transfer (ICT) excited state in a polar environment[22,23]. Following one-photon excitation up to the optically allowed $S_2$ state, the optically forbidden $S_1$ state and the ICT state are generated[22–26]. The presence of the ICT excited state allows nearly 100% EET from carotenoid to chlorophyll in the LH complexes from algae and diatoms[27–29]. In a previous study, fucoxanthin (see Fig. 2a for its chemical structure) was successfully incorporated into the LH1 complex from *Rsp. rubrum* strain G9+[30]. This was remarkable because the algal carotenoid fucoxanthin has a closed end-ring structure and multiple functional groups (allene and carbonyl moieties) in its molecular framework in striking contrast to the relatively simple linear carotenoids, without end-rings, that are always found in the purple bacterial LH systems. However, the EET efficiency from fucoxanthin to Bchl *a* determined by fluorescence excitation spectroscopy was only 28%. Nevertheless, a femtosecond time-resolved absorption study suggested that the EET efficiency with fucoxanthin in LH1 may be higher than that initially determined because of the presence of a significant pool of inappropriately bound fucoxanthin. In the current investigation, we have improved the reconstitution methods and have been successful in incorporating a higher plant carotenoid β-apo-8′-carotenal (see Fig. 2a) into the LH1 complex from *Rsp. rubrum* strain G9+. The EET efficiency from β-apo-8′-carotenal to Bchl *a* was determined to be 77–79% by fluorescence excitation spectroscopy, almost the highest carotenoid to Bchl *a* EET seen in this system so far. Femtosecond time-resolved absorption measurements clearly show that the involvement of the ICT excited state of β-apo-8′-carotenal plays the key role in achieving this efficient EET.

## Results and discussion
### Reconstitution of β-apo-8′-carotenal into the LH1 complex from *Rsp. rubrum* strain G9+. 
Figure 2b shows the steady-state absorption spectra of the LH1 complex from the carotenoidless mutant *Rsp. rubrum* strain G9 + together with the reconstituted LH1 complex with β-apo-8′-carotenal (hereafter abbreviated as Reβapo) and β-apo-8′-carotenal. Reβapo shows an absorption

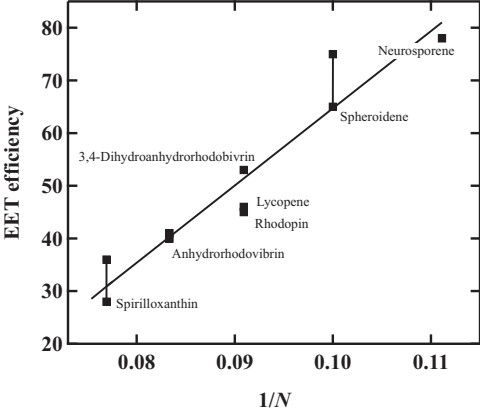

**Fig. 1 The relationship between the efficiency of excitation energy transfer from carotenoid to Bchl *a* in the LH1 complex and the number of conjugated double bonds of the carotenoids.** The linear relationship can be seen between the efficiency of excitation energy transfer from carotenoid to Bchl *a* in the LH1 complex from *Rsp. rubrum* and the inverse of the number of conjugated double bonds (1/$N$) of the carotenoids[18,19].

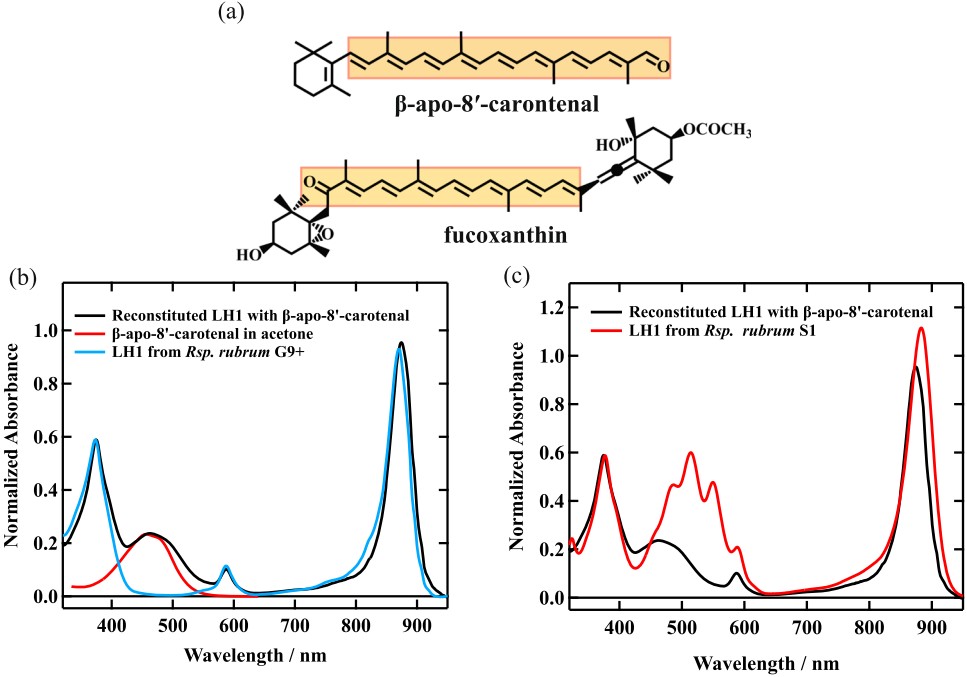

**Fig. 2 Comparison of the steady-state absorption spectra. a** Chemical structures of β-apo-8′-carotenal and fucoxanthin. Conjugated polyene parts coupled with carbonyl group are highlighted with yellow color. **b** Steady-state absorption spectra of Reβapo in 20 mM Tris-HCl buffer (pH 8.0) + 0.58% β-OG (solid black line), β-apo-8′-carotenal in acetone (solid red line), and the LH1 complex isolated from *Rsp. rubrum* strain G9+ in 50 mM phosphate buffer (pH 7.0) + 0.58% β-OG + 10 mM sodium ascorbate (solid blue line). **c** Steady-state absorption spectra of Reβapo in 20 mM Tris-HCl buffer (pH 8.0) + 0.58% β-OG (solid black line), and the LH1 complex isolated from *Rsp. rubrum* strain S1 in 50 mM phosphate buffer (pH 7.0) + 0.6% β-OG + 10 mM sodium ascorbate (solid red line). Absorption intensity is normalized at B880 maximum. All the measurements were carried out at room temperature.

band around 400–600 nm, which suggests that β-apo-8′-carotenal was incorporated into the carotenoidless LH1. The absorption band of β-apo-8′-carotenal in Reβapo is red-shifted by 3 nm from that in acetone (456 nm → 459 nm). This red-shift is caused by the binding of β-apo-8′-carotenal to LH1 protein. Reβapo has its $Q_y$ absorption maximum at 876 nm. This is shifted to the red by 6 nm in comparison to the carotenoidless LH1 complex (solid blue-line in Fig. 2b). The shift of the $Q_y$ absorption maximum reflects the interaction between β-apo-8′-carotenal and B880 Bchl $a$[31]. Previously this shift in the $Q_y$ position has been suggested to reflect the electrostatic interaction between the reconstituted carotenoids and B880 Bchl $a$[18].

Comparison of the intensity of absorbance of the carotenoid absorption region of the native LH1 complex from *Rsp. rubrum* strain S1 (native LH1) and Reβapo shows that in Reβapo it has about half the value seen in the native LH1 (Fig. 2c). If we assume that the molar extinction coefficient (ε) of β-apo-8′-carotenal in LH1 is the same as that of spirilloxanthin ($N = 13$), it can be estimated that half of the carotenoid binding site is reconstituted with β-apo-8′-carotenal in Reβapo. However, the ε value of β-apo-8′-carotenal is expected to be smaller than that of spirilloxanthin, since the ε of β-apo-8′-carotenal in light petroleum is reported to be 109,800 mol⁻¹ cm⁻¹ while that of spirilloxanthin in benzene is reported to be 147,200 mol⁻¹ cm⁻¹ [32,33]. Therefore, the above estimate probably underestimates the number of the reconstituted carotenoids in Reβapo. Nevertheless, we can safely conclude that at least half of the carotenoid binding sites are occupied in Reβapo. We are not sure whether the more efficient incorporation achieved in the case of β-apo-8′-carotenal as compared to the previously studied fucoxanthin[30]. This is because the further purification protocol was applied in the case of Reβapo. In other words, in the case of fucoxanthin reconstituted LH1 there remains the possibility that the excess amount of fucoxanthin is bound inappropriately to LH1.

**Fluorescence, fluorescence excitation, and 1−T spectra of reconstituted LH1.** The EET from carotenoid to B880 Bchl $a$ takes place from both the $S_2$ and $S_1$ states of the carotenoids[22]. In the case of native LH1 complex from *Rsp. rubrum* strain S1, this process is dominated by the EET from the $S_2$ state of the carotenoids to the $Q_x$ state of B880 Bchl $a$[13–17]. Additional EET from the $S_1$ state of carotenoids takes place in the LH1 complexes in which shorter conjugated carotenoids were reconstituted[19]. These LH1 complexes with the additional EET pathway show, therefore, higher efficiency of EET from the carotenoids to B880 Bchl $a$ (see Table 1). In order to determine the EET efficiency from β-apo-8′-carotenal to B880 Bchl $a$ in the present reconstituted LH1 complex, fluorescence emission, and excitation spectroscopic measurements were performed. Figure 3 shows the fluorescence emission, fluorescence excitation, and 1−T spectra of Reβapo, where T is transmittance. The fluorescence emission spectrum was normalized at its intensity maximum. The intense fluorescence signal is observed peaking at 895 nm (solid green line) when exciting at 500 nm, where β-apo-8′-carotenal primarily absorbs excitation light. Both the fluorescence excitation spectrum monitored at 920 nm and 1−T spectra were normalized at the $Q_y$ absorption band of B880 Bchl $a$. The intensities at the $Q_x$ and Soret band positions show the coincidence between the fluorescence excitation and 1−T spectra. This shows that the light energy captured by $Q_x$ or Soret band is transferred to the $Q_y$ band with 100% efficiency: No backward EET is present from the $Q_x$ state to the $S_1$/ICT state of β-apo-8′-carotenal[17]. In the $S_0 \rightarrow S_2$ absorption region of β-apo-8′-carotenal (420–550 nm), the fluorescence excitation spectrum (solid red line) is less intense than the corresponding region 1−T spectrum (solid black line). Presence of the peak of the carotenoid absorption region in the fluorescence excitation spectrum supports that the EET from β-apo-8′-carotenal to B880 Bchl $a$ is taking place. The efficiency of EET from β-apo-8′-carotenal to B880 Bchl $a$ can be calculated

**Table 1 Summary of the EET efficiency from carotenoid to B880 Bchl *a* in the native and carotenoid-reconstituted LH1 from *Rps. rubrum*.**

| Carotenoid | N | EET (%) | Reference |
|---|---|---|---|
| Native | 13 | 27 | 18 |
| Spirilloxanthin | 13 | 28 (36) | 18,19 |
| Anhydrorhodovibrin | 12 | 41 (40) | 18,19 |
| Rhodopin | 11 | 45 | 18 |
| 3,4-Dihydroanhydrorhodovibrin | 11 | 53 | 33 |
| Lycopene | 11 | 45 | 19 |
| Spheroidene | 10 | 65 (75) | 18,19 |
| Neurosporene | 9 | 78 | 19 |
| Fucoxanthin | 7 + C=O + C=·=C | 28 | 30 |
| **β-apo-8′-carotenal** | **9 + C=O** | **79 (77\*)** | This study |

Bold values highlight and distinguish the result of this study from previous studies.
The EET efficiency has been determined by calculating the ratio of fluorescence excitation spectrum divided by 1–T (Transmission) spectrum in carotenoid absorption region. The value denoted with asterisk was determined after eliminating the Bchl *a* absorption in carotenoid absorption region (see Supplementary Note 1).

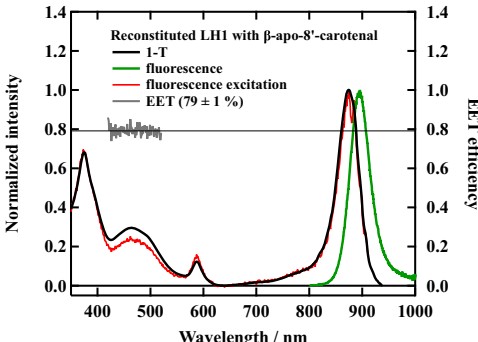

**Fig. 3 Comparison of the steady-state absorption, fluorescence, and fluorescence excitation spectra of Reβapo.** Steady-state absorption spectrum (solid black line) displayed as 1–T (Transmission), fluorescence spectrum excited at 500 nm (solid green line), fluorescence excitation spectrum monitored at 920 nm (solid red line), and the efficiency of EET from β-apo-8′-carotenal to B880 Bchl *a* (solid gray line) are shown for Reβapo in 20 mM Tris-HCl buffer (pH 8.0) + 0.58% β-OG.

by the ratio of the fluorescence excitation spectrum divided by the 1–T spectrum in the carotenoid absorption region (430–550 nm) as shown with the solid gray line in Fig. 3. The average value of the efficiency of EET is 79 ± 1%, which is significantly higher than that from spirilloxanthin in the wild-type S1 strain of *Rsp. rubrum* (27%) and is almost as high as neurosoprene reconstituted LH1 (78%), which shows the highest value among a series of bacterial carotenoids were incorporated (see Table 1). Note that the 79 ± 1% EET efficiency of β-apo-8′-carotenal reconstituted LH1 slightly decreases to 77 ± 1% when the absorption of Bchl *a* in carotenoid absorption region is eliminated (see Fig. S1 in Supplementary Note 1). In this way, we have succeeded in creating a reconstituted LH1 complex with the significantly high efficiency of EET from carotenoid to B880 Bchl *a*.

**Femtosecond time-resolved absorption spectra of the reconstituted LH1.** As shown in Fig. 4b, the femtosecond time-resolved absorption spectrum of β-apo-8′-carotenal recorded at 0.48 ps after excitation in acetone shows a relatively sharp transient absorption band that is ascribable to the $S_1 \rightarrow S_n$ transition in 530–580 nm spectral region. It also shows a broad transient absorption that is ascribable to the ICT excited state in 600–700 nm spectral region, which shows good agreement with the previous studies[34–37]. Therefore, we tentatively assign these transient absorption bands to the $S_1$/ICT states, since the attribution of specific features to $S_1$ or ICT photo-induced absorption is still under debate[22,23]. The similar

$S_1$/ICT characteristic transient absorption spectra are also observed in Reβapo recorded at the delay times of 0.5 and 2.0 ps after excitation (see Fig. 4a). Therefore, it is clear that the ICT excited state is generated in Reβapo following excitation. As shown in Fig. 4a, the peak of the '$S_1 \rightarrow S_n$' like absorption band around 550 nm shifts to blue and decays over time. The ratio of the intensities of transient absorption bands originating from the $S_1$ and ICT states of β-apo-8′-carotenal in Reβapo is similar to those in acetone (Fig. 4b). This suggests that the environment in the binding site of β-apo-8′-carotenal in Reβapo as in acetone has a polar character (see Supplementary Note 2 about the discussion on the biding-site of β-apo-8′-carotenal in Reβapo). The near-infrared spectral region transient absorption spectra shown in Fig. 4a reflect the photo-induced absorbance change of the $Q_y$ absorption band of B880 Bchl *a*. The bleaching signal of the $Q_y$ absorption band is observed immediately after exciting β-apo-8′-carotenal, the fact which suggests the presence of EET from the $S_2$ and/or $S_1$/ICT state of β-apo-8′-carotenal to B880 Bchl *a*. This is because β-apo-8′-carotenal is excited predominantly for femtosecond time-resolved absorption measurements, and hence the immediate bleaching of the $Q_y$ absorption band of B880 Bchl *a* cannot take place without the fast EET from β-apo-8′-carotenal to B880 Bchl *a*.

**Investigation of the relaxation process of the ICT excited states using global and target analyses.** In order to get deeper insight into the excited state relaxations and EET processes in Reβapo, global, and target analyses were performed with the complete datasets of the femtosecond time-resolved absorption spectra using a Glotaran program[38,39]. Although the time-resolved absorption spectral measurements were performed independently in the visible (416–718 nm) and near-infrared (767–1036 nm) spectral regions, the spectral analyses were performed simultaneously in these spectral regions. To begin with, the results in acetone are shown in Fig. 5 (non-normalized EADS are shown in Fig. S3 in Supplementary Note 3). The femtosecond time-resolved absorption spectra in the visible and near-infrared regions are well accounted for using a sequential three components model that have lifetimes of 130 ± 10 fs, 620 ± 10 fs, and 15.4 ± 0.01 ps. The first 130 fs component (solid black line in Fig. 5) can be assigned to the $S_2 \rightarrow S_n$ transition of β-apo-8′-carotenal in acetone by reference to a previous report[34]. The negative peak redder than 550 nm is ascribable for the ground-state bleaching signal[34]. Sharp peaks observed at 580 and 820 nm are due to coherent artifact. The second 620 fs component (solid red line in Fig. 5) is ascribable to the vibrationally hot $S_1$/ICT state, while the third 15.4 ps component (solid blue line in Fig. 5) is ascribable to the $S_1$/ICT state of β-apo-8′-carotenal in acetone

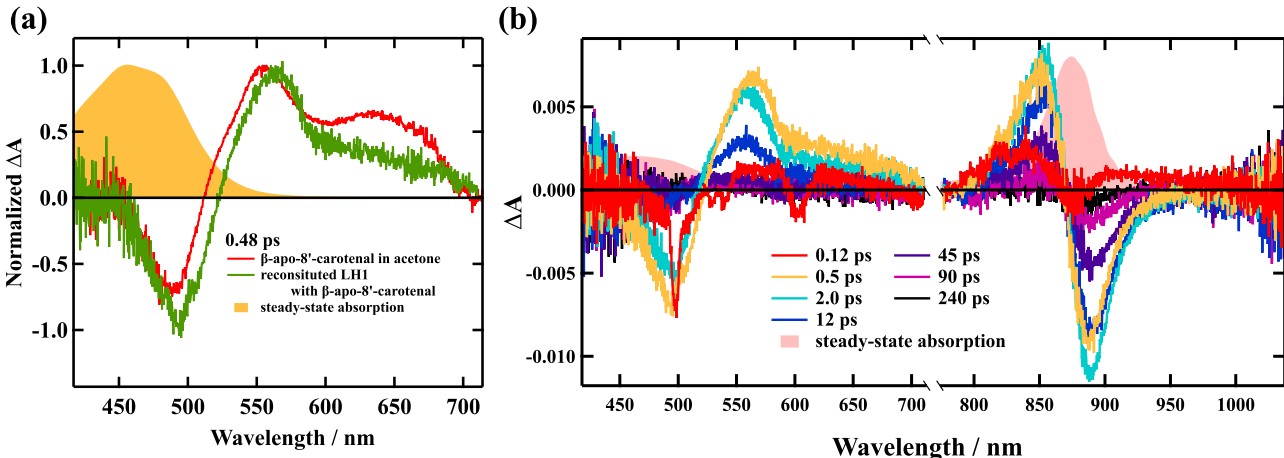

**Fig. 4 Femtosecond time-resolved absorption spectra of β-apo-8′-carotenal in acetone and Reβapo. a** Femtosecond time-resolved absorption spectra of β-apo-8′-carotenal in acetone (solid red line) and in Reβapo (solid green line) recorded at 0.48 ps after excitation. **b** Femtosecond time-resolved absorption spectra of Reβapo in 20 mM Tris-HCl buffer (pH 8.0) + 0.58% β-OG at room temperature following excitation up to the $S_2$ state of β-apo-8′-carotenal. The steady-state absorption spectra of β-apo-8′-carotenal in acetone and that of Reβapo are also shown for comparison. The excitation laser pulse was 490 nm for β-apo-8′-carotenal in acetone and 500 nm for Reβapo. All the measurements were carried out at room temperature.

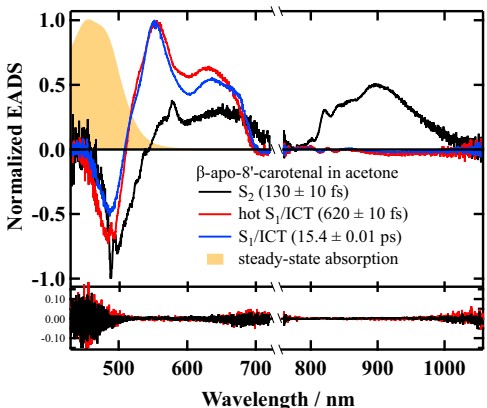

**Fig. 5 EADS obtained by the global analysis against the entire observed dataset of femtosecond time-resolved absorption spectra of β-apo-8′-carotenal in acetone.** The results of global analysis based on a sequential model for β-apo-8′-carotenal in acetone in the visible and near-infrared spectral regions when excited at 490 nm are shown. The first (solid black line) and second (solid red line) right singular value vectors of the residual matrix, which corresponds to the residuals of the fittings, is shown at the bottom. Steady-state absorption spectrum of β-apo-8′-carotenal in acetone is also shown for comparison.

again by reference to the previous report[37]. It is noteworthy here to point out that the spectral pattern of the 15.4 ps EADS (Evolutionally Associated Difference Absorption Spectrum) is composed of a relatively sharp peak around 550 nm, which can be ascribed to the 'S$_1$ → S$_n$' like transition, and a broad absorption in the 600–700 nm spectral region, which has been used as the evidence of the generation of the ICT excited state. Therefore, we have tentatively assigned that this 15.4 ps component is originated from a mixed $S_1$/ICT state[34], although the exact assignment of the $S_1$/ICT state is still under debate[25]. The weak negative spectral features above 850 nm of the 620 fs and 15.4 ps components are attributed to the stimulated emission, which is the good support that these two components have ICT character[24].

The femtosecond time-resolved absorption spectra of Reβapo in the visible and near-infrared spectral regions were successfully

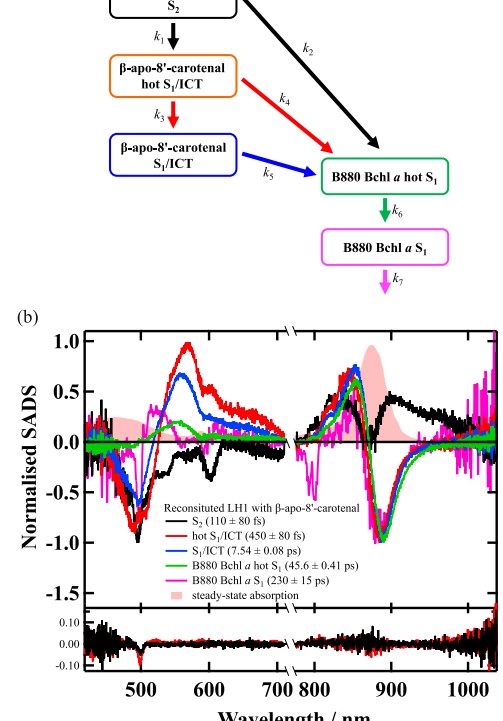

**Fig. 6 Target model and SADS obtained by the target analysis against the entire observed dataset of femtosecond time-resolved absorption spectra of Reβapo. a** A target model used to analyze the femtosecond time-resolved absorption spectra of Reβapo in the visible and near-infrared spectral region. **b** The result of target analysis applied to the entire observed dataset of femtosecond time-resolved absorption spectra of Reβapo. The first (solid black line) and second (solid red line) right singular value vectors of the residual matrix, which corresponds to the residuals of the fittings, is shown at the bottom. The negative peak around 500 nm in the residuals is due to the scattering of the excitation light. Apart from this peak no meaningful spectral pattern is observed in the residuals, a fact which supports the successful spectral fittings. The steady-state absorption spectrum of Reβapo is also shown for comparison.

**Table 2 The rate constants determined by the target analysis using the entire dataset of femtosecond time-resolved absorption spectra of Reβapo in the visible and near-infrared regions based on a target model shown in Fig. 6a.**

|        | Rate |
|--------|------|
| $k_1$  | $(130 \pm 10 \text{ fs})^{-1}$ |
| $k_2$  | $(800 \pm 80 \text{ fs})^{-1}$ |
| $k_3$  | $(620 \pm 10 \text{ fs})^{-1}$ |
| $k_4$  | $(1.67 \pm 0.08 \text{ ps})^{-1}$ |
| $k_5$  | $(7.54 \pm 0.08 \text{ ps})^{-1}$ |
| $k_6$  | $(45.6 \pm 0.41 \text{ ps})^{-1}$ |
| $k_7$  | $(230 \pm 15 \text{ ps})^{-1}$ |

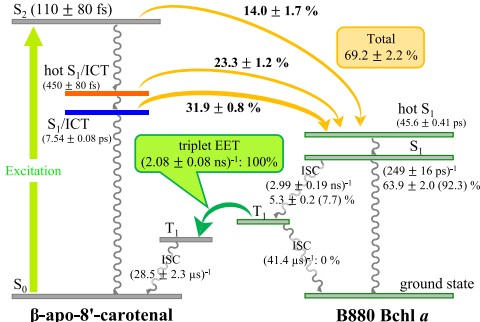

**Fig. 7 Schematic illustration of EET and energy dissipation pathways in Reβapo.** Denoted values indicate rates and overall efficiencies (quantum yields). The rate of each process is expressed with the inverse of lifetime. ISC stands for intersystem crossing.

analyzed based on a target model shown in Fig. 6a. $k_1$ and $k_3$ show the radiative and non-radiative relaxation rates from the $S_2$ and hot $S_1$/ICT states of β-apo-8′-carotenal, respectively. Therefore, we assumed that $k_1 = (130 \text{ fs})^{-1}$ and $k_3 = (620 \text{ fs})^{-1}$ by referring to the results in acetone solution. $k_2$ and $k_4$ show the rates of EET from the $S_2$ and hot $S_1$/ICT states of β-apo-8′-carotenal to B880 Bchl $a$ in Reβapo, respectively. The $S_2$ state energy of β-apo-8′-carotenal should be transferred to the $S_2$ ($Q_x$) state of B880 Bchl $a$, and the $S_2$ B880 Bchl $a$ shows subsequent relaxation to the $S_1$ B880 Bchl $a$. However, we could not incorporate this process in our target model, since the $S_2$ to $S_1$ relaxation rate of Bchl $a$ is as short as 50 fs[40]. $k_5$ shows the rate of the relaxation from $S_1$/ICT state of β-apo-8′-carotenal in Reβapo, which includes both the non-radiative and radiative relaxation rate and the rate of EET from $S_1$/ICT state to B880 Bchl $a$. $k_6$ and $k_7$ show the non-radiative and radiative relaxation rates from the hot $S_1$ and $S_1$ states of B880 Bchl $a$ in Reβapo. The $k_7$ rate constant was fixed to the value determined by sub-nanosecond time-resolved absorption spectroscopy shown later. The result of the target analysis is shown in Fig. 6b and the determined rate constants are summarized in Table 2 (see Fig. S5 in Supplementary Note 5 for the evaluation of the spectral fitting).

Figure 6b shows five components of SADS (Species Associated Difference Absorption Spectra) obtained by global fitting using the above-mentioned target model (the non-normalized SADS are shown in Fig. S4 in Supplementary Note 4). The first component of the SADS ($110 \pm 80$ fs component, solid black-line) is due to the $S_2$ state of β-apo-8′-carotenal in Reβapo. The negative spectral feature in the visible region is mostly due to the ground-state bleaching signal except the sharp peak at 600 nm. This sharp peak might be due to the coherent artifact such as stimulated Raman signal from the solvent water. The positive spectral feature in the near-infrared spectral region is due to transient absorption of B880 Bchl $a$, which shows good evidence that the EET takes place from the $S_2$ state of β-apo-8′-carotenal to B880 Bchl $a$ in Reβapo.

The second SADS ($450 \pm 80$ fs component, solid red-line) and the third SADS ($7.54 \pm 0.08$ ps component, solid blue-line) in Fig. 6b show the characteristic absorption bands that are ascribable to the '$S_1 \rightarrow S_n$' like transition of β-apo-8′-carotenal, respectively, at 575 nm and 550 nm, together with the ICT absorption in 600–700 nm spectral region. Therefore, both SADS are thought to be due to the $S_1$/ICT state. Since the second SADS shows a maximum at a longer wavelength position than the third SADS, we assign the second SADS to the vibrationally hot $S_1$/ICT state of β-apo-8′-carotenal in Reβapo, while the third SADS to the $S_1$/ICT state of β-apo-8′-carotenal in Reβapo by referring to the assignment in acetone solution. The $S_1$/ICT state lifetime in Reβapo is shorter than reported $S_1$ lifetime (26.2 ps[15] and 26 ps[17]) of β-apo-8′-carotenal in non-polar solvents. This lifetime

shortening is caused by the fact the ICT character is expressed when β-apo-8′-carotenal is bound to the LH1 protein: namely, β-apo-8′-carotenal is in the polar environment when it is bound to Reβapo (see the discussion on the binding site of β-apo-8′-carotenal in Reβapo in Supplementary Note 2). Previous reports on peridinin and fucoxanthin when they are bound to Peridinin-Chl $a$-Protein and Fucoxanthin-Chl $a/c$-Protein, respectively, also show similar results[27,29,41]. Moreover, overall spectral band shapes of the second and the third SADS are similar to the $S_1$/ICT transient absorption band of β-apo-8′-carotenal in acetone (see the second EADS in Fig. 5a). Therefore, it is suggested that the LH1 bound β-apo-8′-carotenal is in similar polar environment as in acetone.

The fourth ($45.6 \pm 0.41$ ps) and the fifth ($230.4 \pm 14.9$ ps) SADS in Fig. 6b are ascribable to the vibrationally hot $S_1$ and $S_1$ states of B880 Bchl $a$ by referring to the previous report[42]. It is interesting to note that the more long-lived triplet component of β-apo-8′-carotenal was not clearly detected in Reβapo in this time regime. This may reflect the efficient singlet-singlet EET from β-apo-8′-carotenal to B880 Bchl $a$. It is reported that the formation of triplet carotenoid is suppressed when carotenoids that show highly efficient singlet-singlet EET to Bchl $a$ are reconstituted to the LH system[43].

The singlet-singlet EET pathways from β-apo-8′-carotenal to B880 BChl $a$ in Reβapo is summarized in Fig. 7. The lifetimes of the $S_2$, hot $S_1$/ICT and $S_1$/ICT species of β-apo-8′-carotenal in Reβapo are shorter than those in acetone. This finding reflects EET from β-apo-8′-carotenal to B880 Bchl $a$ in Reβapo. The efficiency of EET can be calculated from the lifetimes determined by global and target analyses; the following Eq. (1) was used to calculate the efficiency of singlet-singlet EET ($\Phi_{\text{EET}}$) from β-apo-8′-carotenal to B880 Bchl $a$.

$$\Phi_{\text{EET}} = \frac{k_{\text{EET}}}{k_R + k_{\text{EET}}} \times 100 (\%) \qquad (1)$$

Here, $k_{\text{EET}}$ is the rate of EET in Reβapo. $k_R$ is the rate of the non-radiative and radiative relaxations of β-apo-8′-carotenal in acetone.

$k_R$ and $k_{\text{EET}}$ of the $S_2$ state of β-apo-8′-carotenal in Reβapo were determined to be $(130 \pm 10 \text{ fs})^{-1}$ and $(800 \pm 80 \text{ fs})^{-1}$, respectively. Therefore, the efficiency of EET from the $S_2$ β-apo-8′-carotenal to $S_2$ ($Q_x$) B880 Bchl $a$ was calculated to be $14.0 \pm 1.7\%$. $k_R$ and $k_{\text{EET}}$ of the hot $S_1$/ICT state of β-apo-8′-carotenal in Reβapo were determined to be $(620 \pm 10 \text{ fs})^{-1}$ and $(1.67 \pm 0.08 \text{ ps})^{-1}$, respectively. Therefore, the efficiency of EET from the hot $S_1$/ICT β-apo-8′-carotenal to $S_1$ ($Q_y$) B880 Bchl $a$ was calculated to be $27.1 \pm 1.4\%$. $k_R$ of the $S_1$/ICT state of β-apo-8′-carotenal in acetone was determined to be $(15.4 \pm 0.01 \text{ ps})^{-1}$,

while the relaxation rate ($k_R + k_{EET}$) of the $S_1$/ICT β-apo-8′-carotenal in Reβapo was determined to be $(7.54 \pm 0.08 \text{ ps})^{-1}$ (see $k_5$ in Table 2). Therefore, the efficiency of EET from the $S_1$/ICT β-apo-8′-carotenal to $S_1$ ($Q_y$) B880 Bchl $a$ was calculated to be $50.9 \pm 1.2\%$. Using these values of the EET efficiency, total efficiency of EET from β-apo-8′-carotenal to B880 Bchl $a$ can be calculated to be $69.2 \pm 2.2\%$, which agrees somewhat well with that determined by fluorescence excitation spectroscopy ($79 \pm 1\%$ or $77 \pm 1\%$), although the complete agreement could not be achieved and nearly 10% discrepancy can be seen unfortunately. However, the most striking finding in this study is that the $S_1$/ICT state becomes the major EET channel in Reβapo. This idea is supported by the recent publication by Tumbarello et al.[44]

As shown in the Introduction section, the number of conjugated C=C bonds ($N$) of open-chain linear carotenoids and the efficiency of EET from carotenoid to Bchl $a$ in the LH1 has a linear relationship. However, how to count $N$ of β-apo-8′-carotenal is not straightforward since β-apo-8′-carotenal has carbonyl group (C=O) conjugated to the polyene backbone. In this case, it has been claimed that the effective number of conjugated double bonds ($N_{eff}$) can be calculated from the 0-0 transition energy of the steady-state absorption spectrum of carotenoids in $n$-hexane solution[45]. The maximum wavelength of the 0-0 transition of β-apo-8′-carotenal in $n$-hexane is 481 nm (20,790 cm$^{-1}$ in energy), which corresponds to $N_{eff} = 9.8$ (see Fig. S6 in Supplementary Note 6). Compared to the linear carotenoid spheroidene ($N = 10$), $N_{eff}$ of β-apo-8′-carotenal is slightly shorter. The efficiency of EET from spheroidene to B880 Bchl $a$ is reported to be 65–75%[18,19]. The total EET efficiency from β-apo-8′-carotenal to B880 Bchl $a$ was determined to be $79 \pm 1\%$ ($77 \pm 1\%$), which is slightly higher than the case of spheroidene. This result can be somehow accounted for if we adopt the $N_{eff}$ value of β-apo-8′-carotenal, which is slightly smaller than that of spheroidene ($N = 10$). However, the efficiency of EET from neurosporene ($N = 9$) is reported to be 78%, the amount of which is almost equivalent to the case of β-apo-8′-carotenal. This remarkable result cannot be explained based on the simple consideration of $N_{eff}$ of β-apo-8′-carotenal, since $N_{eff}$ of β-apo-8′-carotenal is larger than $N$ of neurosporene (see Fig. 8).

The EET from $S_1$/ICT was found to be dominant when compared to the EET efficiency from $S_2$. This finding is quite unusual in the case of bacterial light-harvesting systems, although similar trend has already been found in the LH complexes from algae and diatoms. The slow onset of $Q_y$ bleaching signal of B880

Bchl $a$, found also in the present study, support the EET pathway from both the hot $S_1$/ICT and $S_1$/ICT sate of β-apo-8′-carotenal to the $Q_y$ state of B880 Bchl $a$ being present (see Fig. S7 in Supplementary Note 7). This trend is clearly different from the previous reports for the LH1 complex in which the EET from the $S_2$ state of carotenoids to the $Q_x$ state of B880 Bchl $a$ is the dominant one[13–17]. The predominant EET pathway from $S_1$/ICT to $Q_y$ is presumably due to the coupling of the ICT excited state to the $S_1$ state. Recently, Marcolin et al. reported that the ICT state is strongly bound to the $S_2$ state in fucoxanthin in polar organic solvents[46]. This finding is consistent with our previous results using Stark absorption spectroscopy[47]. However, the present observation is different from those of Marcolin et al. because the 0-0 band of the $S_0$–$S_2$ transition is clearly photoexcited in this study; Macrolin et al. excited the long wavelength edge of the $S_0 \rightarrow S_2$ transition of fucoxanthin because of the limitation of the pump spectrum. In other words, we can propose that the $S_1$ state, which is originally an additional EET channel of the EET to B880 Bchl $a$, is elevated to a major EET channel by the coupling with the ICT excited state to form $S_1$/ICT[27].

There is an opposite view that claims the ICT state has little effect on the EET efficiency based on a study on LH1 from a different species of photosynthetic bacterium that contains spheroidenone (a carbonyl containing carotenoid) as a major carotenoid[48]. However, our present study clearly shows that this may not always be true.

**Sub-nanosecond time-resolved absorption spectroscopy on Reβapo.** Finally, we would like to address whether β-apo-8′-carotenal in Reβapo is photoprotective or not. It is always important to make sure the presence of the photoprotective function of carotenoid in the reconstituted LH system in order to guarantee the robustness of the system. To this end, sub-nanosecond time-resolved absorption spectroscopy has been applied to Reβapo. Figure 9 shows the results of the global analysis based on a 3-components sequential model using the entire observed dataset of the sub-nanosecond time-resolved absorption spectra of Reβapo. The first EADS ($230 \pm 15$ ps component) shown with solid black-line in Fig. 9 can be assigned to the $S_1$ state of B880 Bchl $a$ judging from its lifetime. The second EADS ($2.08 \pm 0.08$ ns component) shown with solid red-line in Fig. 9

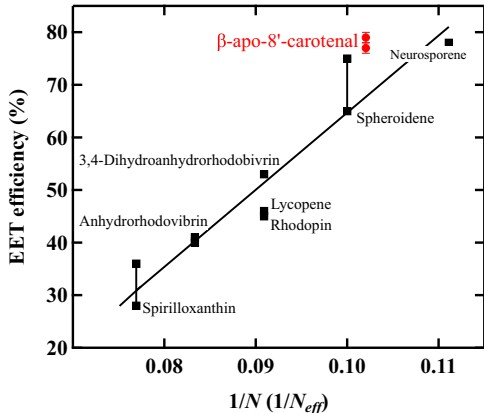

**Fig. 8 The relationship between the efficiency of EET from β-apo-8′-carotenal to Bchl $a$ in Reβapo and the effective number of conjugated double bonds.** The relationship between the efficiency of EET from β-apo-8′-carotenal to Bchl $a$ in Reβapo and the inverse of the effective number of conjugated double bonds ($1/N_{eff}$) is overlaid on the plot shown in Fig. 1.

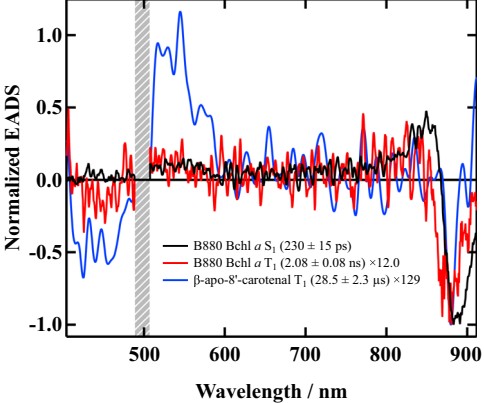

**Fig. 9 EADS obtained by the global analysis against the entire observed dataset of sub-nanosecond time-resolved absorption spectra of Reβapo.** EADS obtained by the global analysis against the entire observed dataset of nanosecond time-resolved absorption spectra of Reβapo based on a 3-components sequential model are shown. The hatched bar at 500 nm shows the region of pump laser light. The spectra were truncated at this pump laser spectral region to reduce the effect of the pump light scattering.

can be assigned to the $T_1$ state of B880 Bchl $a$ judging from both the spectral shape and the lifetime[42]. The third EADS (29.7 µs component) shown with solid blue-line can undoubtedly be assigned to the $T_1$ state of carotenoid in Reβapo judging from the spectral peak around 530 nm and the lifetime[42], although the spectral amplitude is very low and hence the spectrum becomes noisy.

As shown in Fig. 7, the singlet excitation-energy transferred from β-apo-8′-carotenal to B880 Bchl $a$ in Reβapo is dissipated from the $Q_y$ state of B880 Bchl $a$ both radiatively and non-radiatively. During this dissipation process, a part of the $Q_y$ ($S_1$) state of B880 Bchl $a$ transforms to the $T_1$ state by intersystem crossing. How much of the $S_1$ B880 Bchl $a$ transforms to the $T_1$ state can be calculated using the amplitude of the first and second EADS shown in Fig. 9 (see Supplementary Note 8 for the details of the calculation). It turned out that the 7.7% of the $S_1$ B880 Bchl $a$ transforms to the $T_1$ B880 Bchl $a$, and 92.3% of the $S_1$ energy of B880 Bchl $a$ is dissipated by radiative and non-radiative relaxation. It is noteworthy here that this 7.7% efficiency of triplet-triplet energy transfer shows good agreement with the result (8%) of the LH2 complex from *Rhodobacter sphaeroides* strain 2.4.1[49]. This $T_1$ B880 Bchl $a$ decays with the rate of $(2.08 \pm 0.08 \text{ ns})^{-1}$ through the processes of both triplet-triplet EET to β-apo-8′-carotenal and intersystem crossing. The $T_1$ β-apo-8′-carotenal dissipates its triplet energy harmlessly as heat to the surrounding environment with the rate of $(28.5 \pm 2.3 \text{ µs})^{-1}$. According to our previous study on the carotenoidless LH1 complex from *Rsp. rubrum* strain G9+[42], the rate of the intersystem crossing of the $T_1$ B880 Bchl $a$ to go back to the ground state was determined to be $(41.4 \text{ µs})^{-1}$. Therefore, the efficiency ($\Phi_{\text{EET}}^{\text{T}}$) of triplet-triplet energy transfer from $T_1$ B880 Bchl $a$ to β-apo-8′-carotenal can be calculated according to the following Eq. (2).

$$\Phi_{\text{EET}}^{\text{T}} = \frac{k_{\text{EET}}^{\text{T}}}{k_{\text{ISC}} + k_{\text{EET}}^{\text{T}}} \times 100(\%) \qquad (2)$$

Here, $k_{\text{ISC}}$ is the rate of intersystem crossing of $T_1$ B880 Bchl $a$ ($k_{\text{ISC}} = (41.4 \text{ µs})^{-1}$) and $k_{\text{EET}}^{\text{T}}$ is the rate of triplet-triplet energy transfer from $T_1$ B880 Bchl $a$ to β-apo-8′-carotenal. Since $k_{\text{ISC}} + k_{\text{EET}}^{\text{T}} = (2.08 \pm 0.08 \text{ ns})^{-1}$ has been determined by sub-nanosecond time-resolved absorption spectroscopy, $\Phi_{\text{EET}}^{\text{T}}$ can be calculated to be 100%. Therefore, this result suggests that photoprotection by β-apo-8′-carotenal in Reβapo could be functional in Reβapo.

## Conclusion
The present study provides new insights into the role of the ICT excited states of carotenoids by successfully incorporating a higher plant carotenoid, β-apo-8′-carotenal, to the carotenoidless LH1 system from a purple photosynthetic bacterium, *Rsp. rubrum* strain G9+. Importantly, the EET from the $S_1$/ICT state to B880 Bchl $a$ is the major channel in Reβapo. It is clear that the ICT state plays a key role in the realization of highly efficient EET from carotenoid to Bchl $a$. Reβapo can be robust since the photoprotective mechanism of β-apo-8′-carotenal seems to be functional. The present study clearly suggests a strategy of how to go beyond the currently available EET efficiency from carotenoid to Bchl $a$ by harnessing the properties of ICT states.

## Experimental section
**Cell growth and preparation of chromatophores**. Cells of *Rsp. rubrum* strain G9+ (a carotenoidless mutant) were photosynthetically grown under anaerobic conditions with C-succinate medium modified from Cohen-Bazire et al. at 27 °C[50]. The cells were harvested by centrifugation ($18,800 \times g \times 10$ min at 4 °C),

resuspended in 20 mM MES buffer (pH 7.0), and stored in a freezer (−30 °C) until use. Chromatophores of *Rsp. rubrum* strain G9+ were prepared as described previously[51].

**Isolation and purification of B820 heterodimers from the native LH1 complex from *Rsp. rubrum* strain G9+**. The native LH1 complex without carotenoid was isolated from the freshly prepared chromatophores of *Rsp. rubrum* strain G9+ according to the reported protocols[52,53]. The B820 heterodimers were isolated from the native LH1 complex as described previously[54–56] following solubilization with 1.2% octyl-β-glucopyranoside (β-OG) in 50 mM phosphate buffer (pH 7.0) with the presence of 10 mM sodium ascorbate. The B820 heterodimers were further purified by means of sucrose density gradient (0.6–0.8 M) ultracentrifugation ($165,000 \times g \times 16$ h at 4 °C) and stored in the freezer (−30 °C) until required.

**Reconstitution of β-apo-8′-carotenal into the LH1 complex**. The reconstitution of carotenoid into the LH1 complex was performed using the reported method by Yukihira et al.[30] with a slight modification. All-*trans*-β-apo-8′-carotenal were purchased from Sigma-Aldrich and used without further purification. The acetone solution of β-apo-8′-carotenal ($OD_{456} = 1.2$) was very slowly dropped, under vigorous flow of $N_2$ gas, into a 50 mM phosphate buffer (pH 7.0) solution of B820 ($OD_{820} = 6$–8) containing 1.2% β-OG at room temperature. β-apo-8′-carotenal was added until its absorbance band became three times as large as that of the $Q_x$ absorption band of Bchl $a$. The solution was then diluted with the same volume of 50 mM phosphate buffer (pH 7.0) with 10 mM sodium ascorbate so that the final concentration of β-OG becomes 0.6%. This procedure (dilution of the detergent concentration) induced the reassembly of the B820 heterodimers to form the LH1 ring structure[53]. During this process β-apo-8′-carotenal was incorporated into the LH1 complex. Following the reconstitution of the LH1 complex which contains excess amount of carotenoid, the reconstituted LH1 complex was further purified by ion-exchange chromatography, using DEAE-cellulose (DE52, Whatman) as a fixed phase. The reconstituted LH1 complex was charged to the DE52 column (1.5 cm i.d.) pre-equilibrated by 20 mM Tris-HCl buffer (pH 8.0) containing 0.58% β-OG. Excess carotenoids were eluted in 20 mM Tris-HCl buffer (pH 8.0) containing 0.58% β-OG and 50 mM NaCl, and the reconstituted LH1 complex was eluted using 20 mM Tris-HCl buffer (pH 8.0) containing 0.58% β-OG and 500 mM NaCl.

**Steady-state spectroscopic measurements**. Steady-state absorption spectral measurements were performed using a JASCO V-730 UV-vis spectrophotometer. Fluorescence emission and excitation spectra of the reconstituted LH1 complexes ($OD_{876} = 0.3$) were recorded with a HORIBA Duetta spectrofluorometer using a 1 cm optical path-length quartz cuvette at room temperature. The fluorescence-excitation spectra were detected at 920 nm which corresponds to the emission from the B880 absorption band. The efficiency of EET from β-apo-8′-carotenal to Bchl $a$ in the LH1 complexes was determined by comparing the fluorescence excitation spectra with the fractional absorption (1−T) spectra normalizing at the peak of the $Q_y$ transition of B880 Bchl $a$. The 1−T spectra were measured by HORIBA Duetta for the purpose of accurate comparison of the fluorescence excitation and 1−T spectra.

**Femtosecond time-resolved absorption measurements**. A part of the output pulses from a femtosecond Ti:Sapphire regenerative amplifier (Spectra Physics, Solstice Ace, 80 fs pulse duration, 3.5 mJ/pulse (3.5 W), 1 kHz repetition) were guided to excite an

optical parametric amplifier (Spectra Physics, TOPAS-Prime) and a wavelength converter (Spectra Physics, NirUVis) to generate pump pulses. The excitation pulse at 490 or 500 nm used to excite the $S_0 \to S_2$ absorption band of β-apo-8′-carotenal was generated with this set-up. The excitation intensity was set to 20 nJ/pulse and the beam was focused on to the sample with a diameter of 200 μm. Another small part of the output from a femtosecond Ti:Sapphire regenerative amplifier (80 fs pulse duration, 100 μJ/pulse (100 mW), 1 kHz repetition) were guided to a sapphire plate after passing through a rotational neutral density filter to generate probe super-continuum pulses using a nonlinear optical process of self-phase modulation. A sapphire plate of 2 mm thickness was used to generate the probe pulses when recording the transient absorption spectra in the visible spectral region from 430 to 720 nm, while that with 7 mm thickness is used when recording the transient absorption spectra in the near-infrared spectral region from 770 to 1036 nm. The diameter of the probe beam was 250 μm at the sample position. To avoid unnecessary confusion caused by the anisotropy, we chose the magic angle (54.7°) configuration for the polarization between the pump and probe pulses. The probe pulses were irradiated to the sample with 1 kHz repetition rate, while the repetition rate of the pump pulses was reduced to 500 Hz using an optical chopper (Terahertz Technologies Inc., C-995 Optical Chopper) synchronously operated with the 1 kHz output signal from the oscillator of the femtosecond regenerative amplifier. The probe pulses that passed through the sample is collected by a spectrometer (Acton Research Corporation, SpectraPro 275) after passing through a Glan-Thompson prism and the spectral dataset was recorded by a 1024-channel multichannel photodiode detector (Hamamatsu, Linear image sensor S3903-1024Q mounted on linear image sensor driving circuit C7884). The time-resolution of the present set-up is estimated to be around 110 fs by the cross-correlation of pump and probe pulses supposing both of which have 80 fs pulse duration $(80 \times \sqrt{2} = 113\,\text{fs})$, although the time-width of the instrumental response function determined by the global analysis is around 60 fs. The accuracy of the steps of the delay-time stage is <10 fs, and the shortest time interval when recording the time-resolved absorption spectra was 20 fs. Therefore, when we estimate the standard errors of the rate constants by global analysis, we have adopted 10 fs as the standard error if the estimated error value becomes smaller than 10 fs by the global analysis.

The sample was placed in a static quartz cuvette that had an optical path-length of 2 mm. The solution inside the cuvette was continuously stirred using a micro stirrer bar during the measurements. The optical density of the sample was set to be 0.2–0.3 at the excitation wavelength. The integrity of the sample was checked by comparison of the steady-state absorption spectra of the sample before and after the measurements. All the time-resolved measurements were performed at room temperature.

**Sub-nanosecond time-resolved absorption measurements.** Sub-nanosecond pump-probe spectroscopic measurements were performed using a combination of a Ti:Sapphire regenerative amplifier (Hurricane-X, Spectra Physics, 100 fs pulse duration at 800 nm) and a sub-ns pump-probe time-resolved absorption spectrophotometry system (EOS Vis-Nir, Ultrafast Systems)[49,57]. Excitation pulses at 500 nm were prepared through the second harmonic generation of the output of an optical parametric amplifier (OPA-800 CF, Spectra Physics). The excitation intensity was set to 20 nJ/pulse with a pulse width of 100 fs and a beam diameter of 200 μm. The time delay between the pump and probe pulses was adjusted electronically (Ultrafast Systems, EOS). All the measurements were performed at room temperature. The instrumental response function of the system was estimated to be ~500 ps, which corresponded

to the pulse width of the probe super-continuum light; however, the pump light was as short as 100 fs.

## Data availability
The data that support the findings of this study are available from the authors on reasonable request.

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

## Acknowledgements

N.Y. thanks the financial support from JSPS KAKENHI for JSPS fellowship for junior researcher (No. 20J10152). H.H. thanks JSPS KAKENHI, Grant-in-Aids for Basic Research (B) (No. 16H04181) for financial support. H.H. and D.K. are grateful to JSPS KAKENHI, Scientific Research on Innovative Areas "Innovations for Light-Energy Conversion (I⁴LEC)" (No. 17H06433, No. 17H06437 & No. 18H05173) by JSPS for financial support. A.T.G. was funded by the Czech Science Foundation project Photo-Gemm+ 19-28778. R.J.C. gratefully acknowledges support from the Photosynthetic Antenna Research Center, an Energy Frontier Research Center funded by the U.S. Department of Energy, Office of Science, Office of Basic Energy Sciences under Award Number DE-SC 0001035.

## Author contributions

N.Y. made all the samples, performed the steady-state spectroscopy measurements, analyzed the time-resolved absorption data, and drafted the original manuscript. C.U. performed all the time-resolved measurements and analyzed the data. K.H. and D.K. manipulated the time-resolved absorption measurements and contributed to set up the time-resolved absorption spectroscopy system. A.T.G. supervised the sample preparation. R.J.C. contributed to supervise the sample preparation and interpretation of the results and revised the manuscript. H.H. conceptualized this work, supervised the work of N.Y., K.H., and C.U., contributed to analyze the spectral data, and revised the original and revised drafts of the manuscript. All authors contributed to revise the manuscript.

## Competing interests

The authors declare no competing interests.
