## [Peer Review File · Communications Chemistry]

Reviewers' comments:

Reviewer #1 (Remarks to the Author):

Carotenoids typically serve as accessory light-harvesters in photosynthetic complexes. In this work, the authors incorporate the higher plant carotenoid β -apo-8'-carotenal into the bacterial photosynthetic light-harvesting protein LH1 and show improved light harvesting capabilities relative to the native pigment composition. Building on previous work incorporating fucoxanthin into the carotenoidless LH1 strain, *Rhodospirillum rubrum* G9+ (Reference 30), the authors' use of β -apo-8'-carotenal allows for better incorporation and higher energy transfer capabilities to chlorophyll, as established through time-resolved and steady state methods. Furthermore, the authors assign energy transfer through an intramolecular charge-transfer (ICT) state, which further enhances energy transfer relative to predictive models based on the conjugation length. While this study will be of interest to members of the community interested in natural and artificial light-harvesting, several points must be addressed prior to publication related to the accuracy of the model used and claims of photoprotection, as outlined below.

Major Comments:

- 1) On page 7, it is noted that the binding site of β -apo-8'-carotenal in Re β apo is polar in character because it supports an ICT, however the authors should expand upon this assignment and identify binding sites in LH1, and comment on how this is similar (or different) compared to LH1 variants with native carotenoids. Furthermore, the authors should offer a physical mechanism for the more efficient incorporation seen in β -apo-8'-carotenal as compared to the previously studied fucoxanthin (Ref 30).
- 2) The authors should assign the negative features in the region redder than 550 nm in Figure 5a for the S2 state, as one would expect the bleach to be closer to 500 nm (in line with reference 35). Furthermore, it is unclear why the EADS appears structured with peaks at 540 and 570 nm.
- 3) The authors should assign the 23.8 ps component in Figure 5b. It could be helpful to plot the non-normalized EADS such that the approximate weighting of all components can be seen (for example, it is unclear how much of the signal is comprised of this 23.8 ps component in 5b).
- 4) While the chlorophyll bleach band appears within the first ~ 0.5 ps after photoexcitation in the transient absorption spectra for Re β apo (Figure 4a), the global model (Figure 6) used to recover the energy transfer efficiencies of each state (Figure 7) does not take this data into account. The authors' claims of energy transfer pathways would be better supported if the NIR bands associated with energy transfer from the carotenoid to the chlorophyll were also included in the global model. Furthermore, the global timescales of energy transfer to chlorophyll are 1.2 and 12 ps (Figure 6), while minimal changes are seen in the chlorophyll bleach bands after 0.5 ps in the raw TA spectra (Figure 4a), thus including these NIR signatures in a global model could help refine the accuracy of these energy transfer processes.
- 5) With regards to their global model of Re β apo, the authors should discuss physically (1) why the red-ICT state transfers energy more efficiently than the blue-ICT state and (2) why a hot ICT state is not observed in Re β apo like that seen in β -apo-8'-carotenal in acetone.
- 6) While the efficient energy transfer to the triplet state is intriguing, the indirect nature of this measurement and lack of substantiation of biological photoprotection, makes the statement on page 15, "Therefore, this clear result shows that photoprotection by β -apo-8'-carotenal in Re β apo is fully functional" somewhat overstated.
- 7) References for assignment of the carotenoid and chlorophyll triplet spectral features in Figure 9 should be given when discussed in the text on page 14.
- 8) On page 12, the authors report, "Here, the contribution of the radiative rate constant is omitted since the fluorescence of β -apo-8'-carotenal is negligible", however the fluorescence quantum yield for β -apo-8'-carotenal is not reported or cited.
- 9) The time resolution derived for the transient absorption experiments should be shown, along with respective error bars, as it is currently unclear if the 0.08 and 0.09 ps components are artifactual, and if they are not, whether these two timescales corresponding to the S2 decay for Re β apo and β -apo-8'-

carotenal are differentiable with the experimental set-up.

Minor Comments

(1) Table 1 "fucoxanthin" line should be reference 30 not 29

(2) Typo on page 13, "The EET from S1/ICT was found to be dominant when compared to the EET efficiency from S2 "

Reviewer #2 (Remarks to the Author):

In this paper, the authors report an interesting study on the role of carotenoids' (almost) dark ICT state on the EET taking place during light harvesting.

They incorporated an ICT carotenoid into LH1 and verified that this led to improved EET efficiency from the carotenoid to the BChla with respect to the wild-type LH1.

The paper presents solid evidence, and the data seem to support the final conclusions. Overall, I suggest publication with some revisions.

My main criticism is about the general tone of the paper. Most of the treated topics are still highly controversial and several interpretations are typically available. Nonetheless, the majority of these topics are taken for granted (for example the existence of a red and blue form of ICT carotenoids, the attribution of specific features to S1 or ICT photoinduced absorption, the effective relationship between the number of double bonds in carotenoid's structure and EET efficiency,...) .

I suggest reconsidering the text highlighting the still provisional nature of the interpretations and the presence of debate in the literature.

Figure 3 and EET efficiency determination.

Why use 1-T rather than the more immediate absorbance?

The EET efficiency has been calculated by comparing the 1-T and fluorescence spectrum. The difference is very small. To what extent can the authors be sure that this is a significant difference not due to experimental conditions or artifacts?

Along the same line, I have some concerns about the reliability of the final quantitative results. The authors extract quantitative information on the energy flow dynamics (efficiency and time constant for each separate EET pathway). Nonetheless, it is not clear what is the estimate of the experimental error (10%, 20%??) nor the role of the effective percentage of carotenoids effectively occupying the binding sites. I suggest revising the discussion with a more careful quantification of the experimental uncertainty.

I basically agree with the main conclusions and with the qualitative concepts expressed in figure 7. A recent paper [doi: 10.3390/ijms23095067] reported a similar picture and it supports the final claims. Several EET channels from carotenoids to chlorophylls are found and the importance of ICT and red states in the efficiency of the overall energy transport is highlighted. However, I do not fully believe in the numbers provided in Figures 7 and 8. I suggest discussing more the limitations/errors connected with the provided estimates of yields and time constants.

Reviewer #3 (Remarks to the Author):

The authors prepared novel reconstituted LH1 complex by incorporating beta apo 8' carotenal in carotenoidless strain G9+ of *Rhodospirillum rubrum*. They performed steady-state absorption, 1-T,

fluorescence, fluorescence excitation, femtosecond and sub-nanosecond time-resolved absorption measurements. Using global and target analysis they prepared full scheme of excitation energy transfer from S2 state of carotenoid down to ground states of car and B880 BChl a. That's remarkable work because the inserted carotenoid has end-ring, and also contains carbonyl group. Furthermore authors claim that efficiency of energy transfer from carotenoid to BChl a is nearly 80%, which is also remarkable in reconstituted LH1 complexes.

This last statement, though, needs further support. The efficiency was calculated from the ratio of the fluorescence excitation spectrum (of ReBapo) and 1-T spectrum in the carotenoid region. I consider this more precise way of determining efficiency than from lifetimes of S1 or S2 states of carotenoid in solution and in complex. However, the authors didn't take in mind (or did not show that it's negligible) that some part of 1-T spectrum in carotenoid area is also from BChl a. It's hard to judge how much of the spectrum belongs to BChl a from Figure 2a, since it has been normalized. I assume that by comparing 1-T spectrum of carotenoidless LH1 and 1-T spectrum of reconstituted LH1 one can estimate/calculate contribution of BChl a.

If I make rough estimate that BChl a contribution be 10% (which could be wrong and I assume the authors would correct me) then the efficiency of transfer will fall to 77%, which is beyond error margins presented in manuscript. (Let $E/A = \text{eff}$ be efficiency calculated by method in the article, E – emission signal, A – (1-T) signal, $E = E_B + E_C$, E_B emission due direct excitation of BChl a and E_C emission via carotenoid transfer, AB – BChl part and AC carotenoid part of (1-T). Then $\text{eff} = (E_B + E_C) / (AB + AC)$. If we use $E_B / AB = 1$ that is assumption from the manuscript, $AC = 9 * AB$ my estimate and $\text{eff}_C = E_B / AB$ is efficiency attributable only to carotenoid transfer then we get $\text{eff} = (\text{eff}_C * AC + 1/9 * AC) / (AC + 1/9 * AC)$. By cancelling out AC and putting in all the numbers $(\text{eff}_C + 1/9) * 9/10 = 0.8$ we get $\text{eff}_C = 0.88 - 0.11 = 0.77$)

Following analysis of data using global and target analysis is straightforward and nearly schoolbook. Still I wouldn't dare to estimate efficiency of energy transfer from S2 state of carotenoid to Qx of B880. The value of S2 state lifetime in solvent is 0.09 ps and in the LH1 the lifetime is 0.08 ps. I would like to know what are authors' estimated uncertainties of those two values, bearing in mind that laser pulse is around 80 fs, the smallest step (judging from Fig. S3) in the kinetic trace is about 30 fs and signal to noise ratio in Figure 4a.

I suggest the manuscript for publication after clarifying above mentioned subjects. It will undoubtedly be of interest to others in the field.

Responses to Reviewers' comments:

Responses to the comments from Reviewer #1:

*Carotenoids typically serve as accessory light-harvesters in photosynthetic complexes. In this work, the authors incorporate the higher plant carotenoid β -apo-8'-carotenal into the bacterial photosynthetic light-harvesting protein LH1 and show improved light harvesting capabilities relative to the native pigment composition. Building on previous work incorporating fucoxanthin into the carotenoidless LH1 strain, *Rhodospirillum rubrum* G9+ (Reference 30), the authors' use of β -apo-8'-carotenal allows for better incorporation and higher energy transfer capabilities to chlorophyll, as established through time-resolved and steady state methods. Furthermore, the authors assign energy transfer through an intramolecular charge-transfer (ICT) state, which further enhances energy transfer relative to predictive models based on the conjugation length. While this study will be of interest to members of the community interested in natural and artificial light-harvesting, several points must be addressed prior to publication related to the accuracy of the model used and claims of photoprotection, as outlined below.*

Reply: Thank you very much for your accurate and positive evaluation of our study. We have made revisions according to your precious suggestions.

Major Comments:

*1) On page 7, its noted that the binding site of β -apo-8'-carotenal in *Re β apo* is polar in character because it supports an ICT, however the authors should expand upon this assignment and identify binding sites in LH1, and comment on how this is similar (or different) compared to LH1 variants with native carotenoids. Furthermore, the authors should offer a physical mechanism for the more efficient incorporation seen in β -apo-8'-carotenal as compared to the previously studied fucoxanthin (Ref 30).*

Reply: Thank you very much for your valuable comments. We have discussed the environment of the biding site of β -apo-8'-carotenal by referring to the amino-acid sequences of the LH1- α and LH- β polypeptides as well as the PDB data in

Supplementary Information. We are not sure whether the more efficient incorporation achieved in the case of β -apo-8'-carotenal as compared to the previously studied fucoxanthin. This is because the further purification protocol was applied in the case of Re β apo. In other words, in the case of fucoxanthin reconstituted LH1 there remains the possibility that the excess amount of fucoxanthin is bound inappropriately to LH1. We have described this situation in the revised manuscript.

2) The authors should assign the negative features in the region redder than 550 nm in Figure 5a for the S₂ state, as one would expect the bleach to be closer to 500 nm (in line with reference 35). Furthermore, it is unclear why the EADS appears structured with peaks at 540 and 570 nm.

Reply: Thank you very much for the good suggestion. At first, we must apologize that we have made a mistake to describe the excitation wavelength for the time-resolved absorption measurement of β -apo-8'-carotenal in acetone (Fig. 4 and 5). It was 490 nm. We have revised this issue in the revised manuscript. As this reviewer kindly suggested, we have assigned the negative features in the region redder than 550 nm in Figure 5 for the S₂ state along the line with reference 35 in the revised manuscript. We have also added the steady-state absorption spectrum of β -apo-8'-carotenal in acetone for comparison in the revised Fig. 5. The sharp peak at 540 nm in the first EADS (130 fs component) becomes not clear in the revised global analysis using the data both in the visible and near infrared spectral regions simultaneously. The sharp peaks at 580 nm and 820 nm are due to coherent artifact. We have explained this issue in the revised manuscript.

3) The authors should assign the 23.8 ps component in Figure 5b. It could be helpful to plot the non-normalized EADS such that the approximate weighting of all components can be seen (for example, it is unclear how much of the signal is comprised of this 23.8 ps component in 5b).

Reply: Thank you very much for the kind suggestion. According to your thoughtful suggestion shown below, we have re-analyzed the data using both the visible and near infrared spectral regions simultaneously. Then, we found that 23.8 ps component was

not necessary to explain the entire dataset in the visible and near infrared regions. The new analysis afforded us the results of more convincing fittings. Thank you. Also, as this reviewer kindly suggested, we have added the non-weighted (non-normalized) EADS in Supplementary Information. Showing only the non-normalized EADS is not good enough to explain the temporal dependence of the concentration profiles. Therefore, we showed both the non-normalized EADS and the time-dependence of each EADS in the Supplementary Information.

4) While the chlorophyll bleach band appear within the first ~0.5 ps after photoexcitation in the transient absorption spectra for Re β apo (Figure 4a), the global model (Figure 6) used to recover the energy transfer efficiencies of each state (Figure 7) does not take this data into account. The authors' claims of energy transfer pathways would be better supported if the NIR bands associated with energy transfer from the carotenoid to the chlorophyll were also included in the global model. Furthermore, the global timescales of energy transfer to chlorophyll are 1.2 and 12 ps (Figure 6), while minimal changes are seen in the chlorophyll bleach bands after 0.5 ps in the raw TA spectra (Figure 4a), thus including these NIR signatures in a global model could help refine the accuracy of these energy transfer processes.

Reply: Thank you very much for the thoughtful comments. Actually, the excitation energy-transfer from the S₂ β -apo-8'-carotenal to Bchl *a* was included in our global model. We must apologize that our model shown in Fig. 6(b) in the original manuscript was not good enough to convince this fact to the reviewer. We also express our sincere gratitude to the thoughtful suggestion to include the NIR signature in a global model. According to the precious suggestion we were successful to re-analyze the entire dataset of femtosecond time-resolved absorption incorporating simultaneously the visible and NIR data. This new analysis provided us with more accurate and reliable rate constants to calculate the EET efficiency. It is also to be noted that according to this revised analysis we do not have to suppose the presence of the red- and blue-forms of the S₁/ICT state to explain the EET efficiency from β -apo-8'-carotenal to B880 Bchl *a* in Re β apo. Therefore, we have completely revised Fig. 6 and the corresponding discussion in the revised manuscript. Thank you! We have also shown the non-normalized SADS and time-dependence of each SADS in Supplementary Information.

5) *With regards to their global model of Re β apo, the authors should discuss physically (1) why the red-ICT state transfers energy more efficiently than the blue-ICT state and (2) why a hot ICT state is not observed in Re β apo like that seen in β -apo-8'-carotenal in acetone.*

Reply: Thank you very much for the good suggestion. As explained above, we do not have to think the presence of the red- and blue-forms of S₁/ICT states in the revised analysis. We could determine the presence of the hot S₁/ICT state and the EET efficiency from this hot S₁/ICT state to B880 Bchl *a*.

6) *While the efficient energy transfer to the triplet state is intriguing, the indirect nature of this measurement and lack of substantiation of biological photoprotection, makes the statement on page 15, “Therefore, this clear result shows that photoprotection by β -apo-8'-carotenal in Re β apo is fully functional” somewhat overstated.*

Reply: This reviewer's claim is correct. We have revised the sentence more appropriately in the revised manuscript. Thank you.

7) *References for assignment of the carotenoid and chlorophyll triplet spectral features in Figure 9 should be given when discussed in the text on page 14.*

Reply: We have revised as suggested.

8) *On page 12, the authors report, “Here, the contribution of the radiative rate constant is omitted since the fluorescence of β -apo-8'-carotenal is negligible”, however the fluorescence quantum yield for β -apo-8'-carotenal is not reported or cited.*

Reply: We have omitted this statement and analyzed the data using k_R , which shows the non-radiative and radiative relaxation rate.

9) *The time resolution derived for the transient absorption experiments should be shown, along with respective error bars, as it is currently unclear if the 0.08 and 0.09 ps components are artifactual, and if they are not, whether these two timescales*

corresponding to the S₂ decay for Reβapo and β-apo-8'-carotenal are differentiable with the experimental set-up.

Reply: Thank you very much for the important comments. We have clarified the time-resolution of our set-up in the experimental section. We have also added the standard errors to all the rate constants determined in this study. According to the re-analysis using the femtosecond time-resolved absorption data both in the visible and near infrared regions simultaneously, the S₂ state lifetimes of β-apo-8'-carotenal in acetone and in Reβapo were re-determined to be 130 ± 10 fs and 110 ± 10 fs, respectively. These values are beyond the pulse duration (80 fs) of the pump and probe pulses of our femtosecond time-resolved absorption set-up, and the validity of the 20 fs time difference of these two lifetimes can be supported by the estimated standard errors of the measurements and analysis.

Minor Comments:

(1) Table 1 “fucoxanthin” line should be reference 30 not 29

(2) Typo on page 13, “The EET from S₁/ICT was found to be dominant when compared to the EET efficiency from S₂”

Reply: We have revised as has been suggested. Thank you.

Responses to the comments from Reviewer #2:

In this paper, the authors report an interesting study on the role of carotenoids' (almost) dark ICT state on the EET taking place during light harvesting. They incorporated an ICT carotenoid into LH1 and verified that this led to improved EET efficiency from the carotenoid to the BChl a with respect to the wild-type LH1. The paper presents solid evidence, and the data seem to support the final conclusions. Overall, I suggest publication with some revisions.

Reply: Thank you very much for your high evaluation on our study. This encourages us a lot. We have made revisions according to your precious suggestions.

My main criticism is about the general tone of the paper. Most of the treated topics are still highly controversial and several interpretations are typically available. Nonetheless, the majority of these topics are taken for granted (for example the existence of a red and blue form of ICT carotenoids, the attribution of specific features to S₁ or ICT photoinduced absorption, the effective relationship between the number of double bonds in carotenoid's structure and EET efficiency, ...). I suggest reconsidering the text highlighting the still provisional nature of the interpretations and the presence of debate in the literature.

Reply: Thank you very for the precious comments. These comments are right. We have revised the manuscript as this reviewer kindly suggested. We could successfully re-analyze the femtosecond time-resolved absorption dataset without supposing the presence of the red- and blue-forms of the S₁/ICT state. Therefore, we have completely omitted the discussion of this aspect in the revised manuscript (see also the reply to Reviewer #1).

Figure 3 and EET efficiency determination. Why use 1-T rather than the more immediate absorbance? The EET efficiency has been calculated by comparing the 1-T and fluorescence spectrum. The difference is very small. To what extent can the authors be sure that this is a significant difference not due to experimental conditions or artifacts?

Reply: The reason why we need to use 1 – T rather than absorbance is that absorbance is in logarithmic scale of the fraction of the exciting light absorbed by the solution. Since fluorescence intensity linearly depends on the excitation light intensity, the fluorescence excitation spectrum should be compared to the fraction of the exciting light absorbed by the solution (1 – T spectrum), and not to absorbance spectrum. This issue was described in the historical works by Weber and Teale (1957) and Balke and Becker (1968). Indeed, the shape of the fluorescence excitation spectrum is nearly the same with 1 – T spectrum except for the carotenoid absorption region. This is good evidence that the excitation energy-transfer efficiency of carotenoid to Bchl *a* is not unity in Reβapo. This issue is described in Supplementary Information.

- G. Weber and F.W.J. Teale, Fluorescence excitation spectrum of organic compounds in solution, *Trans. Faraday Soc.* **54** (1958) 640-648.
- D.E. Balke and R.S. Becker, Relationship between the Absorption and Excitation Spectra and Relative Quantum Yields of Fluorescence of all-trans-Retinal, *J. Am. Chem. Soc.* **90** (1968) 6710-6711.

Along the same line, I have some concerns about the reliability of the final quantitative results. The authors extract quantitative information on the energy flow dynamics (efficiency and time constant for each separate EET pathway). Nonetheless, it is not clear what is the estimate of the experimental error (10%, 20%??) nor the role of the effective percentage of carotenoids effectively occupying the binding sites. I suggest revising the discussion with a more careful quantification of the experimental uncertainty.

Reply: Thank you very much for the important comments. We have shown standard errors for all the rate constants and efficiencies determined in this study in the revised manuscript.

I basically agree with the main conclusions and with the qualitative concepts expressed in figure 7. A recent paper [doi: 10.3390/ijms23095067] reported a similar picture and it supports the final claims. Several EET channels from carotenoids to chlorophylls are found and the importance of ICT and red states in the efficiency of the overall energy transport is highlighted. However, I do not fully believe in the numbers provided in Figures 7 and 8. I suggest discussing more the limitations/errors connected with the provided estimates of yields and time constants.

Reply: Thank you very much for letting us know the important reference. Fortunately, or unfortunately, it turned out that we do not have to care anything about the red-form of the S₁/ICT state according to the analysis using the visible and near infra-red spectral regions data simultaneously. Therefore, we have fully omitted the part of the discussion on the red and blue forms of the S₁/ICT state in the revised manuscript. Nevertheless, the importance of the involvement of the ICT state to boost the light-harvesting capacity of Reβapo is still intact. Therefore, we have added the citation of this reference. We

have also added the standard errors to all the rate constants and the EET yield determined in this study.

Responses to the comments from Reviewer #3:

The authors prepared novel reconstituted LH1 complex by incorporating beta apo 8' carotenal in carotenoidless strain G9+ of Rhodospirillum rubrum. They performed steady-state absorption, 1-T, fluorescence, fluorescence excitation, femtosecond and sub-nanosecond time-resolved absorption measurements. Using global and target analysis they prepared full scheme of excitation energy transfer from S2 state of carotenoid down to ground states of car and B880 BChl a. That's remarkable work because the inserted carotenoid has end-ring, and also contains carbonyl group. Furthermore authors claim that efficiency of energy transfer from carotenoid to BChl a is nearly 80%, which is also remarkable in reconstituted LH1 complexes.

Reply: Thank you very much for your high evaluation on our work. This encourages us a lot.

This last statement, though, needs further support. The efficiency was calculated from the ratio of the fluorescence excitation spectrum (of ReBapo) and 1-T spectrum in the carotenoid region. I consider this more precise way of determining efficiency than from lifetimes of S1 or S2 states of carotenoid in solution and in complex. However, the authors didn't take in mind (or did not show that it's negligible) that some part of 1-T spectrum in carotenoid area is also from BChl a. It's hard to judge how much of the spectrum belongs to BChl a from Figure 2a, since it has been normalized. I assume that by comparing 1-T spectrum of carotenoidless LH1 and 1-T spectrum of reconstituted LH1 one can estimate/calculate contribution of BChl a. If I make rough estimate that BChl a contribution be 10% (which could be wrong and I assume the authors would correct me) then the efficiency of transfer will fall to 77%, which is beyond error margins presented in manuscript. (Let $E/A=eff$ be efficiency calculated by method in the article, E – emission signal, A – (1-T) signal, $E=EB + EC$, EB emission due direct excitation of BChl a and EC emission via carotenoid transfer, AB – BChl part and AC carotenoid part of (1-T). Then $eff=(EB+EC)/(AB+AC)$. If we use $EB/AB=1$ that is

*assumption from the manuscript, $AC=9*AB$ my estimate and $effC=EB/AB$ is efficiency attributable only to carotenoid transfer then we get $eff=(effC*AC+1/9*AC)/(AC+1/9*AC)$. By cancelling out AC and putting in all the numbers $(effC+1/9)*9/10=0.8$ we get $effC=0.88-0.11=0.77$).*

Reply: Thank you very much for the thoughtful comment. This comment is quite right. We have recalculated the EET efficiency according to your kind suggestion and discussed in Supplementary Information. Thank you.

Following analysis of data using global and target analysis is straightforward and nearly schoolbook. Still I wouldn't dare to estimate efficiency of energy transfer from S2 state of carotenoid to Qx of B880. The value of S2 state lifetime in solvent is 0.09 ps and in the LH1 the lifetime is 0.08 ps. I would like to know what are authors' estimated uncertainties of those two values, bearing in mind that laser pulse is around 80 fs, the smallest step (judging from Fig. S3) in the kinetic trace is about 30 fs and signal to noise ratio in Figure 4a.

Reply: Thank you very much for the important comments. We have clarified the time-resolution of our set-up in the experimental section. We have also added the standard errors to all the rate constants determined in this study. With regard to the issue of the reliability of the difference of the S₂ state lifetimes of β-apo-8'-carotenal in acetone and in Reβapo, these two lifetimes were re-determined to be 130 ± 10 fs and 110 ± 10 fs, respectively according to the re-analysis using the femtosecond time-resolved absorption data both in the visible and near infrared regions simultaneously. These values are beyond the pulse duration (80 fs) of the pump and probe pulses of our femtosecond time-resolved absorption set-up, and the validity of the 20 fs time difference of these two lifetimes can be supported by the estimated standard errors of the measurements and analysis.

I suggest the manuscript for publication after clarifying above mentioned subjects. It will undoubtedly be of interest to others in the field.

Reply: Once again, thank you very much for your encouragement. We have thoroughly revised the manuscript according to your kind suggestion.

REVIEWERS' COMMENTS:

Reviewer #1 (Remarks to the Author):

The authors have satisfactorily addressed my points and improved the manuscript. I now recommend publication.

Reviewer #2 (Remarks to the Author):

As I wrote in my previous report, I still believe this manuscript deserves publication in CommChem. I think the current form is now publishable.

The authors took into account my suggestions for more quantitative and detailed comments on the spectroscopic data and their errors (a similar objection was also raised by another referee). Also, they made modifications so to make even more explicit the focus and the final claims of the paper (for example removing the discussion about red/blue form of carotenoids, which was not fully necessary for the final conclusions of this paper).

All considered I recommend publication in the current form.

Reviewer #3 (Remarks to the Author):

My comments from the review were satisfactorily explained and necessary changes implemented to the manuscript and I recommend the manuscript for publishing.